# Forecasting Foodborne Disease Risk Caused by *Vibrio parahaemolyticus* Using a SARIMAX Model Incorporating Sea Surface Environmental and Climate Factors: Implications for Seafood Safety in Zhejiang, China

**DOI:** 10.3390/foods14101800

**Published:** 2025-05-19

**Authors:** Rong Ma, Ting Liu, Lei Fang, Jiang Chen, Shenjun Yao, Hui Lei, Yu Song

**Affiliations:** 1Zhejiang Provincial Key Laboratory of Urban Wetlands and Regional Change, Hangzhou Normal University, Hangzhou 311121, China; 2021210214012@stu.hznu.edu.cn (R.M.); tingliu_hz@hznu.edu.cn (T.L.); leihui@hznu.edu.cn (H.L.); songyu@hznu.edu.cn (Y.S.); 2Undergraduate Academic Affairs Office, Fudan University, Shanghai 200438, China; 3Department of Environmental Science and Engineering, Fudan University, Shanghai 200438, China; 4Zhejiang Provincial Center for Disease Control and Prevention, Hangzhou 310051, China; jchen@cdc.zj.cn; 5Key Laboratory of Geographic Information Science (Ministry of Education), East China Normal University, Shanghai 200241, China; sjyao@geo.ecnu.edu.cn; 6School of Geographic Sciences, East China Normal University, Shanghai 200241, China

**Keywords:** *Vibrio parahaemolyticus*, risk factors, lag period, SARIMAX prediction model, food safety

## Abstract

*Vibrio parahaemolyticus* is a prevalent pathogen responsible for foodborne diseases in coastal regions. Understanding its dynamic relationship with various meteorological and marine factors is crucial for predicting outbreaks of bacterial foodborne illnesses. This study analyzes the occurrence of *V. parahaemolyticus*-induced foodborne illness in Zhejiang Province, China, from 2014 to 2018, using an 8-day time unit based on the temporal characteristics of marine products. The detection rate of *V. parahaemolyticus* exhibited a distinct cyclical pattern, peaking during the summer months. Meteorological and marine factors showed varying lag effects on the detection of *V. parahaemolyticus*, with specific lag periods as follows: sunshine duration (3 weeks), air temperature (3 weeks), total precipitation (8 weeks), relative humidity (7 weeks), sea surface temperature (1 week), and sea surface salinity (8 weeks). The SARIMAX model, which incorporates both marine and climatic factors, was developed to facilitate short-term forecasts of *V. parahaemolyticus* detection rates in coastal cities. The model’s performance was evaluated, and the actual values consistently fell within the 95% confidence interval of the predicted values, with a mean absolute error (*MAE*) of 0.047, indicating high accuracy. This framework provides both theoretical and practical insights for predicting and preventing future foodborne disease outbreaks. These findings can support food industry stakeholders—such as seafood suppliers, restaurants, regulatory agencies, and healthcare institutions—in anticipating high-risk periods and implementing targeted measures. These include enhancing cold chain management, conducting timely seafood inspections, strengthening cross-contamination controls during seafood processing, dynamically adjusting market surveillance intensity, and improving hygiene practices. In addition, hospitals and local health departments can use the model’s forecasts to allocate medical resources such as beds, medications, and staff in advance to better prepare for seasonal surges in foodborne illness.

## 1. Introduction

Foodborne diseases are illnesses caused by the ingestion of harmful substances, including biological pathogens, through contaminated food sources [1]. These diseases represent a major global public health challenge. According to the World Health Organization (WHO), approximately 600 million people suffer from foodborne illnesses annually, resulting in 420,000 deaths and placing a substantial burden on public health systems and economies worldwide [2]. In China, a survey on the burden of acute gastroenteritis estimates that around 209 million foodborne disease cases occur annually, with acute gastroenteritis being the predominant symptom. This figure excludes non-infectious foodborne illnesses, highlighting the gravity of foodborne diseases as a leading food safety concern in the country [3].

Over recent decades, outbreaks of foodborne diseases in China have been primarily attributed to microbial factors, with *Vibrio parahaemolyticus* and *Salmonella* being the most common culprits [4]. *V. parahaemolyticus*, a Gram-negative, facultative anaerobic bacterium, is commonly found in seawater and marine organisms such as fish, shrimp, and shellfish. Existing data indicate that the main sources of foodborne infections caused by *V. parahaemolyticus* are live seafood, raw/rare seafood, freshwater fish, raw meat, and raw fowl [5]. It is one of the leading pathogens responsible for foodborne illnesses in China’s coastal regions [6]. Due to its halophilic nature, *V. parahaemolyticus* thrives in marine environments, making it a significant public health risk in coastal areas. In Zhejiang Province, where seafood consumption is increasing, understanding the factors contributing to outbreaks is essential for effective prevention and control. Therefore, it is critical to identify the risk factors and assess the impact of foodborne illnesses caused by *V. parahaemolyticus* in this region.

A growing body of research has investigated the epidemiological characteristics of bacterial foodborne diseases [7,8,9] and explored their influencing factors [10,11,12,13]. Studies have indicated a long-term upward trend in *Vibrio* spp. infections in U.S. coastal counties, with increased incidence rates and longer hospitalization durations among high-risk populations [7]. Climate anomalies such as El Niño can alter oceanic conditions, shifting warm waters to higher latitudes and triggering *V. parahaemolyticus* outbreaks in previously unaffected regions [14]. Mirón [11] found that rising temperatures may elevate the risk of foodborne microbial contamination. Kim [12] conducted a correlation analysis revealing a significant positive relationship between temperature, precipitation, humidity, and *V. parahaemolyticus* outbreaks, while a negative correlation was found with daylight hours. Moreover, future climate change scenarios, influenced by varying socioeconomic pathways, may alter the level of foodborne disease risks [15]. These studies predominantly emphasize climate and socio-economic factors, often neglecting the significant role marine environmental factors play in *V. parahaemolyticus* contamination of seafood.

Marine environmental variables such as sea surface temperature, salinity, and chlorophyll concentration are strongly correlated with the incidence of bacterial foodborne diseases, exhibiting varying lag effects depending on the time scale. Fletcher [16] used models to investigate *V. parahaemolyticus* growth at different temperatures, confirming a direct link between temperature and bacterial survival. Harrison et al. [17] analyzed sea surface temperature data from the coastlines of England and Wales, concluding that higher sea temperatures facilitate the growth and survival of *Vibrio* species. Hsiao et al. [18] reported a positive impact of sea temperature and salinity on the incidence of *V. parahaemolyticus* in Taiwan. The effects of rainfall, humidity, and sunlight on *V. parahaemolyticus* infection have shown varied results across different studies, often influenced by regional disparities and the absence of systematic marine factor inclusion. In addition to scientific observations, regulatory responses to these risks vary significantly across countries. While studies in the U.S. and Taiwan emphasize the correlation between environmental conditions and Vibrio outbreaks, their food safety policies differ. The U.S. Food and Drug Administration (FDA) mandates strict post-harvest controls for molluscan shellfish, such as rapid cooling and time-temperature monitoring [19]. Taiwan has implemented early warning systems and a hazard-based classification approach to seafood risk. In contrast, China’s seafood safety infrastructure is undergoing modernization, with progress in cold chain logistics and market supervision, though enforcement challenges remain [20]. Compared to Europe’s comprehensive Rapid Alert System for Food and Feed (RASFF) [21], which facilitates real-time cross-border responses, many developing regions still face policy gaps. These discrepancies underscore the importance of data-driven, locally adaptable models like SARIMAX, which can supplement traditional inspection-based approaches and help authorities prioritize proactive interventions. Moreover, the impact of climate and marine factors extends beyond bacterial growth in the natural environment to various stages of the food supply chain. Elevated ambient and sea temperatures can compromise cold chain integrity during seafood storage and transportation, while increased humidity may affect hygiene standards in seafood markets and processing facilities. These factors collectively heighten the risk of cross-contamination and pathogen amplification, underscoring the need for predictive models that account for environmental conditions across the entire seafood value chain. Recent advancements in time series analysis have been used to examine trends in foodborne disease incidence [22,23,24]. These methods have helped predict disease outbreaks and inform public health policies [25,26,27,28]. For instance, de Noordhout et al. [22] employed various models to predict cases of *Salmonella*, *Campylobacter*, and *Listeria* infections in Belgium from 2012 to 2020. In Melbourne, Australia, a study utilized a distributed lag nonlinear model to examine the short-term relationship between climate conditions, such as temperature and rainfall, and salmonellosis [24]. The findings revealed that temperature exhibited the most robust correlation with the risk of salmonellosis, with a lag of 4 weeks, while rainfall showed no significant association. Park [25] applied seasonal ARIMA modeling to predict the impact of climate on bacterial foodborne disease incidence in hospitalized patients. While these studies have provided valuable insights, most focus on annual or monthly scales, leaving room for more detailed analyses at finer time scales. Liang et al. [28] used the ARIMA and SARIMAX models to identify hotspots for future COVID waves using a dataset of COVID-19 cases. Moreover, they often concentrate on disease incidence alone, overlooking the influence of external environmental variables such as climate and marine factors, which should be incorporated to improve prediction accuracy.

While previous research has typically isolated meteorological or marine factors, this study combines both using time series analysis to predict *V. parahaemolyticus* outbreaks. Using Zhejiang Province as a case study, we analyze foodborne *V. parahaemolyticus* disease spatiotemporal patterns over an 8-day period, or “week”. The study investigates the effects of both marine and meteorological factors on disease incidence and utilizes the SARIMAX model to predict outbreak risks by integrating these environmental factors.

## 2. Study Area and Research Methods

### 2.1. Study Area and Data Sources

Zhejiang Province, located on the southeastern coast of China (see Figure 1), spans an area of 101,800 km^2^, roughly equivalent to the size of Iceland. The province boasts a rugged coastline of 1805 km, the longest among all Chinese provinces. With a highly developed fishing industry, Zhejiang produces over 5 million tons of aquatic products annually, of which more than 4 million tons are marine-based. The province experiences a subtropical monsoon climate, characterized by warm temperatures and abundant precipitation—conditions that are conducive to the growth of microbial pathogens.

Administratively, Zhejiang is divided into 11 prefecture-level cities and has experienced rapid economic development in recent decades. This socio-economic progress has led to increased regional integration and diversification in dietary patterns. As a result, the incidence of foodborne diseases, particularly those caused by *V. parahaemolyticus*, has been relatively high in the region. According to the Zhejiang Foodborne Disease Surveillance System, the detection rate of *V. parahaemolyticus*-related foodborne illnesses has increased in recent years. In response to this growing concern, Zhejiang established its first sentinel hospitals for foodborne disease monitoring in 2010. Currently, 101 sentinel hospitals across 89 districts and counties monitor and report foodborne diseases, with additional sentinel sites located in areas with higher population densities.

The fishing grounds of Zhejiang’s fleet predominantly lie within the geographical coordinates of 20–37° N and 117–128° E. The main fishing activities are concentrated between 26–34° N and 119–128° E, particularly in the Yushan, Wenzhou-Taizhou, Mindong, and Zhoushan fishing grounds. These regions experience the highest fishing intensity and the most frequent annual operations [29]. Zhejiang’s seafood industry comprises a complex network of fishing grounds, aquaculture bases, cold storage facilities, wholesale markets, and distribution logistics. Seafood harvested from marine grounds is typically transported to onshore processing centers before being distributed to urban wholesale markets, restaurants, and retailers. Despite advances in cold chain infrastructure in recent years, seasonal temperature fluctuations and logistical constraints still pose risks of cold chain breaches, which can facilitate the growth of *V. parahaemolyticus*. In rural and coastal areas, informal seafood trade and street vendors are common, often lacking refrigeration or standardized handling protocols. These weak points in the seafood supply chain highlight the need for predictive tools that can support early warning systems and inform strategic interventions during high-risk periods. As these areas are critical to the fishing industry and have high levels of interaction with marine pathogens, they were selected as the focal marine study regions for this research.

### 2.2. Analytical Framework and Data Sources

The analytical framework for this study, based on the acquired and processed multivariate time series data, is illustrated in Figure 2. The analysis proceeded as follows:Lag Correlation Analysis: Initially, a lag correlation analysis was performed to calculate the cross-correlation coefficients between meteorological data, marine satellite products, and *V. parahaemolyticus* detection data at various lag intervals. This step aimed to identify the meteorological and marine environmental factors influencing the detection rate of *V. parahaemolyticus* and to determine the respective lag periods for these influences.Multivariate Time Series Model Construction: Next, a multivariate time series model was developed. The sequential data were subjected to stationarity tests, and non-stationary sequences were differenced to achieve stationarity. The model parameters were identified and optimized using the autocorrelation function (ACF), partial autocorrelation function (PACF), and Bayesian Information Criteria (BIC) to fine-tune the SARIMAX model. The residuals of the model were carefully examined to ensure they adhered to white noise characteristics, which is a crucial assumption for the reliability of the model.Prediction and Evaluation: Finally, the established model was employed to predict the detection rates of *V. parahaemolyticus*, with its performance evaluated through appropriate metrics.

Given the 8-day cycle of ocean satellite products, the meteorological data and pathogen detection rates were aggregated into 8-day intervals for analysis. The specific data sources used in this study are as follows:Meteorological Data: This includes temperature, total precipitation, relative humidity, sunshine duration, and wind speed. These data were sourced from the European Centre for Medium-Range Weather Forecasts (ECMWF) (https://www.ecmwf.int/, accessed on 9 July 2024).Ocean Satellite Products: Sea surface temperature and chlorophyll levels were retrieved from NASA’s ocean data portal (https://oceandata.sci.gsfc.nasa.gov/, accessed on 19 July 2024), while sea surface salinity data were obtained from NASA’s Earth data portal (https://search.earthdata.nasa.gov/search, accessed on 28 July 2024).Foodborne Illness Data: Data on foodborne illnesses caused by *V. parahaemolyticus* were extracted from the Zhejiang Foodborne Disease Surveillance Reporting System. This dataset includes 182,311 cases and corresponding sample test results from 101 sentinel hospitals across the province, covering the years 2014 to 2018. Specific data points include the date, gender, age, address, occupation, and pathogen test results for each case. It should be noted that the dataset used in this study is based on confirmed clinical cases of foodborne illness caused by *V. parahaemolyticus*, as reported through sentinel hospitals. While seafood traceability and contamination testing are occasionally conducted during outbreak investigations, such environmental sampling data were not included in the present analysis. Our model thus focuses on predicting trends in clinical incidence rather than direct contamination levels in seafood products.

### 2.3. Multivariate Time Series Analysis

Time series analysis is a statistical technique used to examine trends, seasonality, periodicity, and randomness within time series data, leveraging these characteristics to forecast future trends. In epidemiology, historical disease data are often utilized to predict future incidence rates. However, incidence rates are influenced not only by historical values but also by other related variables within the same time series, such as sea surface temperature, air temperature, and various climatic and marine factors. To assess the relationship between time series variables, covariance and correlation coefficients are commonly used.

Correlation analysis is a method for identifying interdependencies among variables, quantifying both the strength and direction of their relationships. In the natural sciences, Pearson’s correlation coefficient is widely applied to measure the degree of correlation between two variables [30]. In this study, the Pearson correlation coefficient was employed to examine the lagged correlations between marine and climatic factors and the incidence of bacterial foodborne diseases. The coefficient is defined as the ratio of the estimated sample covariance to the product of the standard deviations of the two variables, and is mathematically expressed as:(1)$$r=\frac{\sum_{i=1}^{n}(X_{i}-X)(Y_{i}-Y)}{\sqrt{\sum_{i=1}^{n}{(X_{i}-X)}^{2}}\sqrt{\sum_{i=1}^{n}{(Y_{i}-Y)}^{2}}}$$
where $X_{i}$ represents the time series variable for marine or climatic factors, and $Y_{i}$ represents the time series variable for the detection rate of *V. parahaemolyticus*; *X* and *Y* are the mean values of the respective time series variables. The correlation coefficient *r* ranges from −1 to +1. When *r* > 0. A positive correlation is indicated by *r* > 0, while a negative correlation is represented by *r* < 0. The closer the absolute value of *r* is to 1, the stronger the correlation. A value of *r* = 0 suggests no linear relationship between the variables. The interpretation of the correlation strength is as follows: a coefficient between 0 and 0.2 indicates no or very weak correlation; 0.2 to 0.4 indicates a weak correlation; 0.4 to 0.6 indicates a moderate correlation; 0.6 to 0.8 suggests a strong correlation; and 0.8 to 1 indicates a very strong correlation.

### 2.4. SARIMAX Model

The Seasonal Autoregressive Integrated Moving Average (SARIMA) model has become a vital tool in public health surveillance and early warning systems for infectious diseases [31,32,33]. Particularly useful for time series forecasting with long-term trends and clear seasonal patterns, the SARIMA model is well-suited for analyzing seasonal and non-seasonal processes in epidemiological data [34]. The general form of the SARIMA model is expressed as SARIMA(*p*,*d*,*q*)(*P*,*D*,*Q*)*s*, where *p* represents the order of non-seasonal autoregression, *d* is the order of non-seasonal differencing, *q* is the order of non-seasonal moving average, *P* represents the order of seasonal autoregression, *D* is the order of seasonal differencing, *Q* is the order of seasonal moving average, and *s* is the length of the seasonal cycle [35]. The mathematical expression of the model is as follows:(2)$$\Phi_{P}\left(B^{s}\right)\phi_{P}\left(B\right){\left(1-B^{s}\right)}^{D}{\left(1-B\right)}^{d}y_{t}=c+\Theta_{Q}\left(B^{S}\right)\theta_{q}\left(B\right)\varepsilon_{t}$$
where $y_{t}$ represents the time series value of the detection rate of *V. parahaemolyticus* at time *t*, $\varepsilon_{t}$ is the white noise sequence at time *t*, and *c* is the constant term. *B* is the backshift operator, where $B^{S}$ denotes shifting $y_{t}$ backward by *s* periods, that is $B^{S}y_{t}$ = $y_{t-s}$. $\phi_{P}\left(B\right)$ and $\theta_{q}\left(B\right)$ are the autoregressive and moving average polynomials of order *p* and *q* respectively. $\Phi_{P}\left(B^{s}\right)$ and $\Theta_{Q}\left(B^{S}\right)$ are the seasonal autoregressive and seasonal moving average polynomials of order *P* and *Q*, with a seasonal period of *s*.

The SARIMAX (Seasonal Autoregressive Integrated Moving Average with Exogenous Variables) model extends the SARIMA framework by incorporating exogenous variables, such as meteorological and marine environmental factors. This extension allows the model to capture not only the periodic characteristics of the disease incidence but also the relationship between the disease and external factors influencing its transmission, such as environmental changes [36].

#### 2.4.1. Stationarity Test

Python 3.6 was utilized to decompose the original time series into trend, seasonal, and residual components. The Augmented Dickey–Fuller (ADF) unit root test was applied to assess the stationarity of the original series. If the series was found to be non-stationary, differencing was performed until the series achieved stationarity. The ADF test was then re-applied to confirm the stationarity of the processed time series. Stationarity tests were conducted on the detection rates of foodborne diseases, meteorological variables, and marine environmental parameters to ensure that all time series met the stationarity condition.

#### 2.4.2. Model Identification and Order Selection

One of the critical steps in constructing a SARIMAX model is determining the optimal model order. The parameters *p*, *d*, *q*, *P*, *D*, *Q*, and *s* are estimated by analyzing the time series plot of the foodborne disease detection rates, along with the autocorrelation function (ACF) and partial autocorrelation function (PACF). After this analysis, a preliminary candidate model is proposed. The Bayesian Information Criterion (BIC) is then computed, and the values of the parameters that minimize the BIC are selected as the optimal model parameters [37]. Using these optimal parameters, a multivariate SARIMAX model is constructed, incorporating climatic and marine environmental factors. For instance, the lag of 3 weeks for temperature represents the typical time required for environmental changes to impact bacterial growth and transmission through seafood.

#### 2.4.3. Model Validation Method

To validate the model, the Ljung–Box Q test was used to assess whether the residuals were white noise. A significance level of α = 5% was chosen. If the test statistic exceeded the critical value $X_{1-\alpha }^{2}\left(m\right)$, the null hypothesis that the residuals were white noise was rejected, suggesting that the model had not fully captured all useful information and needed to be re-fitted [38]. The formula for the Ljung–Box Q test statistic is as follows:(3)$$X^{2}=n\left(n+2\right)\sum_{k=1}^{m}\frac{\eta_{k}^{2}}{n-k}$$
where *n* is the total number of data points in the sequence, and $\eta_{k}^{2}$ is the autocorrelation coefficient of the residual sequence.

#### 2.4.4. Model Prediction Method

The model was trained using data from January 2014 to January 2018, consisting of disease detection rates, climate data, and marine data. The testing data, covering February 2018 to December 2018, was used for model evaluation. Model accuracy was assessed by comparing the predicted values with the observed data.

The relative error between the predicted values and the actual test data provided an initial evaluation of the model’s prediction results. The overall fitting and prediction performance were quantified using the mean absolute error (*MAE*). A lower *MAE* value indicates better model performance in terms of both fitting and prediction accuracy. *MAE* measures the average degree of deviation between model predictions and actual values. The formula for calculating *MAE* is [39]:(4)$$\mathrm{MAE}=\frac{1}{n}\sum_{i=1}^{n}|\hat{y_{i}}-y_{i}|$$
where $\hat{y_{i}}$ is the predicted detection rate of *V. parahaemolyticus*, $y_{i}$ is the actual detection rate, and n is the length of the testing time series.

## 3. Results Analysis

### 3.1. Multivariate Time Series

Between 2014 and 2018, a total of 182,473 bacterial foodborne disease samples were collected in Zhejiang Province, of which 6430 cases (3.52%) tested positive for *V. parahaemolyticus* infection. Notably, 6226 of these cases (approximately 97%) occurred during the summer months, from May to October. Table 1 provides a summary of the descriptive statistics for the 8-day detection rate of *V. parahaemolyticus* and the associated environmental and meteorological conditions on the days of detection, highlighting the overall variation in the dataset.

The time series plot (Figure 3) clearly demonstrates the temporal trends in *V. parahaemolyticus* infections in relation to fluctuations in meteorological and marine conditions. A notable peak in infections occurs annually during the summer, coinciding with increases in both air temperature and sea surface temperature. This pattern suggests a potential correlation between the prevalence of *V. parahaemolyticus* and these environmental factors.

### 3.2. Lagged Correlation Analysis

A correlation analysis was conducted to explore the relationship between the detection rate of *V. parahaemolyticus* infections and various meteorological and marine factors (Figure 4). The analysis revealed significant correlations between *V. parahaemolyticus* detection rates and all five meteorological factors and three marine factors, though with different time lags.

Meteorological Factors: Four meteorological variables showed positive lagged effects on the detection rate of *V. parahaemolyticus*, with varying time lags:
Sunshine duration and air temperature exhibited a lag of 3 weeks;Total precipitation had a lag of 8 weeks;Relative humidity showed a lag of 7 weeks.Marine Factors: Among the marine environmental factors:
Sunshine sea surface temperature demonstrated a positive lagged effect with a lag of 1 week;Sea surface salinity showed a negative lagged effect, with a lag of 8 weeks;Chlorophyll concentration and average wind speed did not show a significant lagged effect on the detection rate. However, a negative correlation was observed between chlorophyll concentration and the detection rate, while average wind speed exhibited a positive correlation.

This lagged correlation analysis suggests that environmental and climatic factors, particularly those associated with temperature and precipitation, may influence the prevalence of *V. parahaemolyticus* infections, with varying time delays. These relationships highlight the potential for environmental monitoring to predict outbreaks of foodborne illness. The identification of lagged environmental drivers—such as air temperature (3 weeks), sea surface temperature (1 week), and relative humidity (7 weeks)—offers actionable insights for food businesses and regulators. For example, the detection of elevated environmental temperatures in early summer may serve as a signal for the seafood industry to reinforce cold chain protocols, including maintaining storage temperatures below 4 °C, the upper safety threshold for perishable seafood products. Moreover, the model’s predictive capacity can inform HACCP planning by identifying high-risk timeframes during which critical control points (CCPs)—such as transportation, market refrigeration, and kitchen handling—require heightened scrutiny. Regulatory agencies may also adjust inspection frequency or intensify compliance monitoring during these windows. By translating environmental surveillance into proactive safety management, our approach supports risk-based decision-making and operational preparedness across the seafood supply chain.

### 3.3. SARIMAX Model Prediction

#### 3.3.1. Stationarity Test of the Time Series

To verify the stationarity of the time series for the detection rate of *V. parahaemolyticus*, as well as for meteorological and marine data, the Augmented Dickey–Fuller (ADF) test was performed. The results of the ADF test, summarized in Table 2, show that the test statistics for all three datasets (disease, meteorological, and marine data) exceed the critical values at the respective significance levels, with *p*-values all less than 0.05. This indicates that the time series for the detection rate of *V. parahaemolyticus* and the associated environmental factors are stationary, satisfying the necessary condition for further analysis.

Following the stationarity test, the characteristics of the time series were further analyzed using the autocorrelation function (ACF) and partial autocorrelation function (PACF), as depicted in Figure 5. The blue—shaded areas in the autocorrelation plot (ACF) and partial autocorrelation plot (PACF) represent the confidence intervals. These intervals are typically used to determine whether the autocorrelation or partial autocorrelation coefficients are significantly different from zero. If a coefficient lies outside the blue—shaded area, it indicates a significant correlation at that lag. The ACF of the original series demonstrated a tapering pattern, indicating a slow decay of correlations. In contrast, the PACF exhibited a cutoff pattern after the first lag, with the values fluctuating around zero within a range of twice the standard deviation. These observations suggest that the time series is appropriately suited for modeling with SARIMAX, as the ACF and PACF patterns are indicative of stationarity and appropriate for further model identification.

#### 3.3.2. Parameter Testing and Model Construction

The preliminary analysis of the parameters in the SARIMAX model for the *V. parahaemolyticus* detection rate time series involved determining the initial values for the model’s components: *p*, *d*, *q*, *P*, *D*, *Q*, and *S*. Since the original data series was found to be stationary, the non-seasonal differencing parameter *d* was set to 0. Given the clear seasonal patterns in the data, which follow an annual cycle, the seasonal differencing parameter *D* was set to 1, and the seasonal period *S* was set to 45.

The autocorrelation function (ACF) of the original series displayed a tapering pattern, while the partial autocorrelation function (PACF) decayed after the first lag. This suggested that the non-seasonal autoregressive order *p* should be set to 1. The non-seasonal moving average order *q* was less clear, so the search range for *q* was set to {0, 1}. Given that the dataset spanned four annual cycles, the search range for the seasonal autoregressive order *P* and seasonal moving average order *Q* was set to {0, 1, 2, 3, 4}.

To refine the model parameters further, the Bayesian Information Criterion (BIC) was used. The BIC helps balance model fit and complexity, with a lower value indicating a better model fit and reducing the risk of overfitting. Using Python 3.6 to evaluate different parameter combinations, the model with the lowest BIC value was SARIMAX(1,0,0)(0,1,1)45, which resulted in a BIC value of 647.88. This set of parameters was selected as the optimal model.

With these optimal parameters, the model incorporated exogenous variables, including meteorological data with lags: sunshine duration lagged by 3 weeks, temperature lagged by 3 weeks, total precipitation lagged by 8 weeks, and relative humidity lagged by 7 weeks. For the marine data, sea surface temperature was lagged by 1 week, chlorophyll concentration had no lag (0 weeks), and sea surface salinity was lagged by 8 weeks. A multivariate SARIMAX model was constructed using these variables as the exogenous factors, providing the optimal model. Table 3 presents the coefficients for each influencing factor within the model, along with their significance levels.

#### 3.3.3. Model Validation

The validation of the SARIMAX model involved analyzing the residuals to assess model adequacy. The Q-Q plot of the residuals (Figure 6a) and the ACF (Figure 6b) and PACF (Figure 6c) plots indicate that the residuals fall within an acceptable margin of error. The red line in subfigure a (Residual Q − Q diagram) is the theoretical line representing the perfect linear relationship between the sample quantiles and theoretical quantiles under the assumption of normality. If the blue dots (representing the actual sample quantiles) closely follow this red line, it indicates that the residuals are approximately normally distributed.The blue—shaded areas in subfigures b (Residual autocorrelation graph) and c (Residual partial autocorrelation plot) represent the confidence intervals. These intervals are used to determine whether the autocorrelation and partial autocorrelation coefficients are significantly different from zero. If a coefficient lies outside the blue—shaded area, it implies a significant correlation at that lag. This suggests that the model effectively captures the key patterns in the data without leaving significant autocorrelated noise.

To further validate the residuals, the Durbin-Watson (D-W) test was performed. The D-W statistic ranges from 0 to 4, where a value near 2 indicates no significant autocorrelation. Values closer to 0 suggest strong positive autocorrelation, while values closer to 4 suggest strong negative autocorrelation. The D-W test result for the SARIMAX model was 1.7755, which is close to 2, indicating no significant autocorrelation in the residuals. This supports the adequacy of the model.

Additionally, the Ljung–Box test was applied to the residuals. With a significance level of 0.05, the *p*-values for the first 20 lags were all greater than 0.05 (Figure 6d), supporting the null hypothesis that the residual autocorrelation coefficients do not significantly differ from zero. This result confirms that the residuals follow a Gaussian white noise pattern. Taken together, these findings indicate that the SARIMAX model fits the data well and is suitable for prediction.

#### 3.3.4. Model Prediction

The SARIMAX model was trained using disease detection rates, climate data, and marine data from January 2014 to January 2018. Data from February 2018 to December 2018 were used as test data to evaluate the model’s predictive performance. The model’s accuracy was assessed by comparing the predicted values to the actual observed values.

The mean absolute error (*MAE*) for the model was 0.047, indicating a high level of prediction accuracy. As shown in Figure 7, the actual observed values fall within the 95% confidence interval of the predicted values. This demonstrates the reliability of the SARIMAX model for predicting *V. parahaemolyticus* foodborne disease outbreaks in Zhejiang Province.

The successful application of the SARIMAX model provides a robust theoretical foundation for the short-term prediction and prevention of foodborne disease outbreaks. This tool can serve as a valuable resource for public health authorities to develop effective interventions and control measures for mitigating the risks associated with *V. parahaemolyticus*.

## 4. Discussion

The detection rate of *V. parahaemolyticus* in Zhejiang Province from 2014 to 2018 was influenced by climate factors, including air temperature, precipitation, relative humidity, and sunshine duration. These factors exhibited varying lag effects on bacterial foodborne disease infections, consistent with the findings of Hsiao et al. [18] in Taiwan and Zhang et al. [40] in Australia. Sunshine duration and air temperature both had a lag of 3 weeks, total precipitation a lag of 8 weeks, and relative humidity a lag of 7 weeks (Figure 4a). Warmer air temperatures, in particular, were positively correlated with the incidence of foodborne diseases, as higher temperatures favor the proliferation of *V. parahaemolyticus*, thereby increasing the likelihood of food contamination [41].

Although average wind speed did not exhibit a lag effect, its potential indirect role in influencing foodborne diseases cannot be overlooked. High wind speeds may facilitate the spread of bacterial contaminants by affecting the reproduction, survival, and persistence of *V. parahaemolyticus* in the environment [42]. Relative humidity and sunshine duration also showed positive lagged effects, altering the survival and transmission patterns of *V. parahaemolyticus* and leading to an increase in bacterial foodborne disease cases. This is consistent with findings from research conducted in South Korea [43]. These results indicate that meteorological factors significantly influence bacterial foodborne disease detection rates, with their lag effects providing valuable insights for disease prediction frameworks. This understanding can be extended to other coastal regions facing similar challenges with bacterial foodborne diseases.

The marine environment in Zhejiang Province further contributes to the risk of *V. parahaemolyticus* foodborne illnesses. The province’s extensive coastline and high seafood consumption expose the population to elevated risks, as *V. parahaemolyticus* thrives in seawater and marine products such as fish, shrimp, and shellfish. Sea surface temperature (SST), chlorophyll concentration, and sea surface salinity exhibit seasonal patterns (Figure 2), with SST closely tracking the disease detection rate, both peaking from June to August, with a 1-week lag. Chlorophyll concentration and sea surface salinity showed opposite trends to the disease detection rate, with chlorophyll concentration having no lag effect and sea surface salinity an 8-week lag (Figure 4b).

Warmer SSTs enhance bacterial growth in seawater, increasing contamination and infection risks. This positive correlation between SST and the disease detection rate aligns with findings from King [44] in New Zealand, Konrad [45] in British Columbia, Canada, and Haley [46] on the Black Sea coast of Georgia. However, the observed negative correlation between chlorophyll concentration and disease detection rate in this study contrasts with results from Urquhart [47], who found a positive correlation in oyster samples from the Great Bay Estuary in New Hampshire. This discrepancy may stem from differences in chlorophyll concentration levels; the concentrations in this study were mostly below 5 µg/L, potentially creating an environment less conducive to bacterial survival. Elevated sea surface salinity also appeared to suppress bacterial growth, consistent with findings from Esteves [48] and Martinez-Urtaza [49].

These results underscore the need for targeted interventions, particularly during the high-risk summer months. Enhanced food safety inspections of seafood and aquatic products, stricter monitoring of restaurants, and improved public awareness of hygiene practices are critical measures. Training staff in seafood markets and restaurants on standard operating procedures can further mitigate infection risks.

Predicting foodborne disease outbreaks is crucial for public health efforts to prevent and control infections [43,50]. The SARIMAX(1,0,0)(0,1,1)45 model, which incorporated lags for key meteorological and marine variables, demonstrated robust predictive performance. With a *MAE* of 0.047, the model accurately captured disease trends during the test period (February to December 2018). The actual values fell within the 95% confidence interval of the predicted values (Figure 7), confirming the reliability of the model for short-term predictions of *V. parahaemolyticus* outbreaks in Zhejiang Province.

While our findings align with previous studies from countries such as the U.S., Taiwan, and Australia in identifying climate-driven patterns in *Vibrio* outbreaks, regional differences in seafood consumption behavior and regulatory infrastructure may affect both exposure risk and mitigation effectiveness. For instance, Zhejiang’s coastal population exhibits a strong preference for fresh and sometimes raw seafood, particularly during the summer, which coincides with peak *V. parahaemolyticus* activity. This differs from inland regions or Western countries, where seafood is more likely to be frozen or cooked thoroughly. Additionally, while regulatory agencies in the EU [21] and U.S. [19] often employ centralized and highly structured food safety alert systems (e.g., RASFF or FDA’s outbreak response protocols), China’s system is still undergoing modernization, with variability in enforcement capacity across regions [20].

Also, this study provides a strong theoretical foundation for public health authorities to implement timely preventive measures. Surveillance efforts should focus on the summer months, when infection risks are highest, and sanitation practices in seafood markets and restaurants should be improved in advance, leveraging the lag times of key influencing factors.

This proactive approach can significantly mitigate the spread of *V. parahaemolyticus* and reduce the burden of foodborne diseases.

For seafood vendors and restaurants, the findings highlight the importance of dynamically adjusting storage practices in response to environmental risk forecasts. For instance, restaurants may reinforce the separation of raw and cooked seafood during predicted high-risk periods, in accordance with zoning principles under HACCP guidelines. Markets may also intensify on-site temperature checks and hygiene inspections as sea and air temperatures rise. From a regulatory perspective, SARIMAX-based predictions can support risk-based supervision by guiding when and where to increase inspection frequency. Regulatory agencies can implement forecast-driven interventions, such as market closures or targeted public advisories, during periods of elevated environmental risk. Furthermore, integrating model outputs into public health communication systems can improve consumer awareness and decision-making. Although this study emphasizes the importance of improving sanitation in seafood markets and restaurants, it should also take into account the unique challenges faced by street food vendors, who may have limited access to refrigeration equipment and proper sanitation facilities. Since informal vendors account for a significant proportion of seafood consumption, predictive models should be used to develop targeted interventions, such as low-cost food safety measures, public awareness campaigns, and adaptive regulatory approaches suitable for street food settings. Future research could explore the effectiveness of these interventions in reducing *V. parahaemolyticus*—related outbreaks in informal food environments. Regarding cold-chain logistics, the findings suggest the need for enhanced infrastructure—particularly mobile or decentralized cold storage hubs that can be activated in anticipation of seasonal outbreaks. For informal or rural seafood markets, local governments may consider supplying temporary refrigeration units or conducting targeted food safety training aligned with seasonal risk profiles. These practical extensions of our modeling framework provide a roadmap for operationalizing predictive analytics within food safety systems at multiple levels. Furthermore, this modeling framework can be adapted for other coastal regions facing similar foodborne disease risks, representing a valuable tool for global public health strategies. Beyond foodborne illness, this framework holds potential for addressing broader issues related to climate and marine environmental changes.

This study acknowledges several limitations and suggests directions for future research:Spatial Variations: This research primarily focused on temporal changes in disease incidence, neglecting spatial variations. Future studies should incorporate spatiotemporal analysis to enhance risk prediction and provide a more comprehensive understanding of disease dynamics.Expanded Predictive Variables: While this study considered meteorological and marine factors, future research should integrate additional variables, such as food consumption patterns, food exposure data, demographic factors (e.g., age structure), and fiscal and healthcare expenditures. Incorporating these factors would enable a more comprehensive understanding of bacterial foodborne disease incidenceGeographical Specificity: This model is set under the conditions of subtropical monsoon climate and highly developed fisheries in Zhejiang Province, China (28–30° N). Not fully applicable to other coastal areas, model parameters need to be modified based on the marine climate environment of other regions.Lack of Direct Seafood Contamination Data: This study relied solely on clinical case data reported by sentinel hospitals. However, foodborne illness originates from pathogen contamination in seafood products, which is not captured in the current dataset. Future research should incorporate direct pathogen testing of seafood samples across various points in the supply chain to better validate temporal associations, identify contamination pathways, and improve the accuracy of early warning systems.

## 5. Conclusions

This study explored the dynamic relationship between marine and meteorological environments in Zhejiang Province and the risk of *V. parahaemolyticus* foodborne disease. By employing the SARIMAX model, this research integrated marine, climate, and autocorrelated disease factors into a predictive framework, offering a novel approach to forecasting foodborne disease outbreaks in coastal regions.

The findings revealed significant temporal clustering of *V. parahaemolyticus* foodborne disease, with peak incidences occurring during the summer months. Key meteorological factors—air temperature, relative humidity, precipitation, sunshine duration, and wind speed—were identified as significant predictors, with lags of 3, 7, 8, 3, and 0 weeks, respectively. Among marine factors, sea surface temperature, chlorophyll concentration, and sea surface salinity were found to be influential, with lags of 1, 0, and 8 weeks, respectively.

Compared with previous studies that primarily modeled meteorological data and disease autocorrelation, this research constructed a SARIMA model and enhanced it by integrating meteorological and marine factors into the SARIMAX model. The optimal SARIMAX(1,0,0)(0,1,1)45 model incorporated lagged predictors, including sunshine duration (3 weeks), air temperature (3 weeks), total precipitation (8 weeks), relative humidity (7 weeks), sea surface temperature (1 week), chlorophyll concentration (0 weeks), and sea surface salinity (8 weeks). Model validation demonstrated that actual disease detection rates fell within the 95% confidence interval of the predicted values, with a mean absolute error (*MAE*) of 0.047, indicating strong predictive accuracy.

This precise determination of lag periods highlights the SARIMAX model’s capability for short-term prediction of *V. parahaemolyticus* foodborne disease outbreaks in Zhejiang Province. The model provides a reliable theoretical foundation for disease prevention and control, while also offering actionable guidance for government departments in Zhejiang to mitigate foodborne disease risks.

By addressing these limitations, future studies can further refine predictive models and improve their applicability to diverse settings, strengthening global efforts to prevent and control foodborne diseases.

## Figures and Tables

**Figure 1 foods-14-01800-f001:**
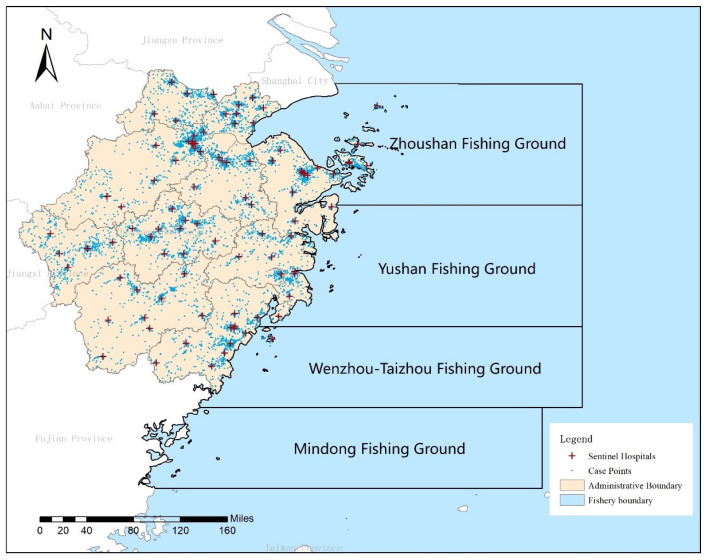
Location of the study area and distribution of positive cases caused by *V. parahaemolyticus*.

**Figure 2 foods-14-01800-f002:**
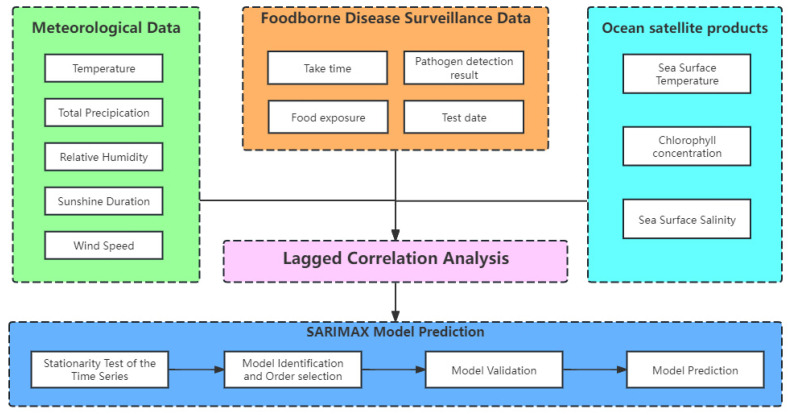
The research framework of our study.

**Figure 3 foods-14-01800-f003:**
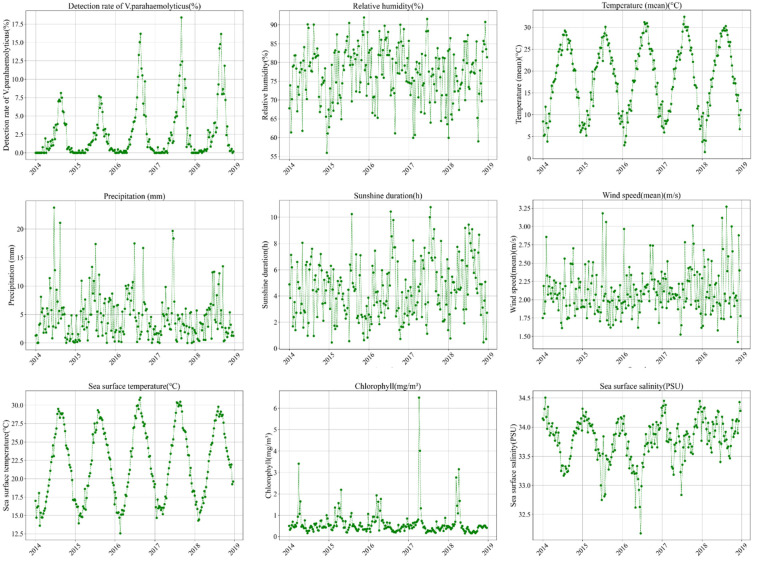
Time series of various parameters.

**Figure 4 foods-14-01800-f004:**
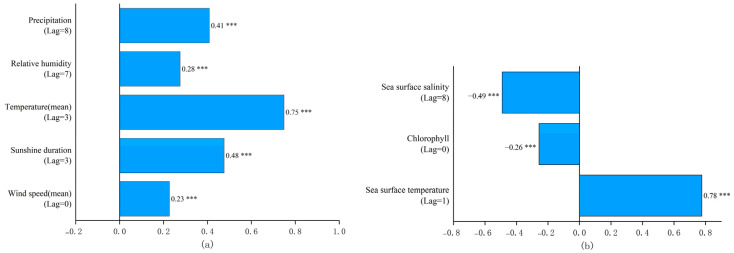
Correlation and lag period between environmental factors and *V. parahaemolyticus* detection rate. (**a**) Climate factors; (**b**) marine factors (Note: *** represents *p* < 0.001).

**Figure 5 foods-14-01800-f005:**
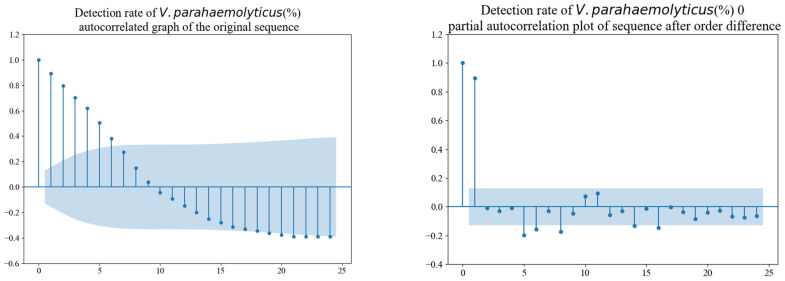
ACF and PACF of the time series.

**Figure 6 foods-14-01800-f006:**
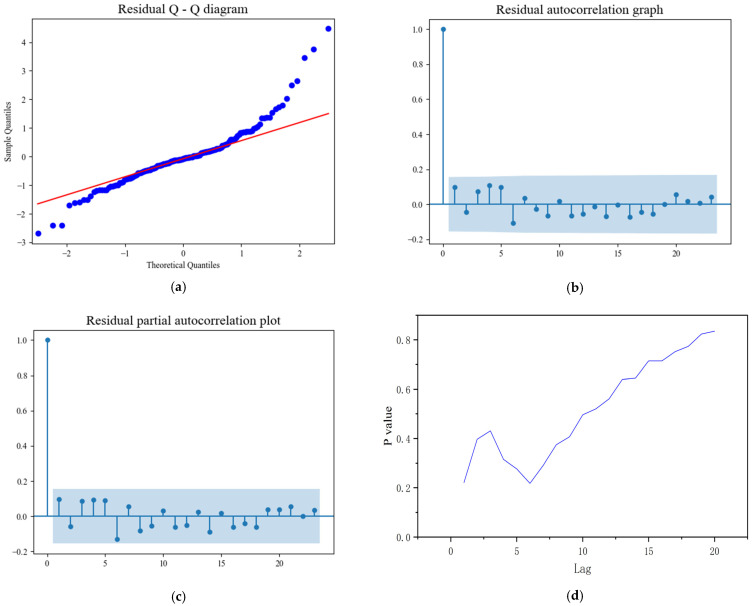
Series plot of the model residuals; (**a**) standardized residual Q-Q plot, (**b**) autocorrelation (ACF) with 5% significance limit, (**c**) partial autocorrelation (PACF) with 5% significance limit, (**d**) Ljung–Box test results.

**Figure 7 foods-14-01800-f007:**
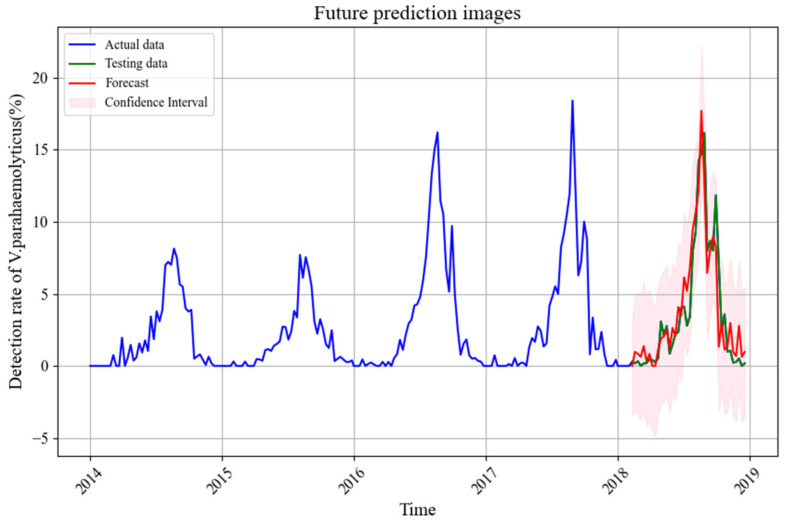
Predicted vs. Actual Values.

**Table 1 foods-14-01800-t001:** Descriptive statistics of variables.

Variable	N	Mean	S.D.	Minimum	Maximum
*V. parahaemolyticus* Detection Rate (%)	229	2.6964	3.7230	0.0000	18.3935
Air Temperature (°C)	229	18.2956	7.9318	1.4378	32.4123
Total Precipitation (mm)	229	4.5351	4.1187	0.0000	23.7683
Relative Humidity (%)	229	77.0814	7.4093	55.9477	91.9691
Sunshine Duration (h)	229	4.4894	2.2546	0.4533	10.7682
Wind Speed (m/s)	229	2.1106	0.3159	1.4226	3.2706
Sea Surface Temperature (°C)	229	22.0112	5.0304	12.5561	31.0386
Chlorophyll Concentration (mg/m^3^)	229	0.6051	0.6296	0.1528	6.4957
Sea Water Salinity (psu,‰)	229	33.7804	0.3810	32.1755	34.5074

**Table 2 foods-14-01800-t002:** ADF stationarity test for time series.

Variable	Test Statistic	*p*-Value	1% Level	5% Level	10% Level
*V. parahaemolyticus* Detection Rate	−4.9433	0.0000	−3.4603	−2.8747	−2.5738
Air Temperature	−7.4811	0.0000	−3.4611	−2.8751	−2.5740
Total Precipitation	−5.4752	0.0000	−3.4598	−2.8745	−2.5737
Relative Humidity	−10.9417	0.0000	−3.4594	−2.8743	−2.5736
Sunshine Duration	−10.3799	0.0000	−3.4594	−2.8743	−2.5736
Wind Speed	−13.6713	0.0000	−3.4594	−2.8743	−2.5736
Sea Surface Temperature	−8.4994	0.0000	−3.4610	−2.8750	−2.5740
Chlorophyll Concentration	−10.8984	0.0000	−3.4594	−2.8743	−2.5736
Sea Water Salinity	−3.5205	0.0075	−3.4596	−2.8744	−2.5736

**Table 3 foods-14-01800-t003:** Coefficients of influencing factors in the model.

Independent Variable	Coefficient	S.E.	z	*p* > |z|	0.025	0.975
Air Temperature	0.0265	0.141	0.188	0.851	−0.249	0.302
Total Precipitation	−0.0302	0.088	−0.364	0.716	−0.193	0.132
Relative Humidity	0.0323	0.046	0.706	0.480	−0.057	0.122
Sunshine Duration	0.1332	0.133	1.005	0.315	−0.127	0.393
Wind Speed	1.1290	0.468	2.411	0.016	0.211	2.047
Sea Surface Temperature	0.2971	0.243	1.224	0.221	−0.179	0.773
Chlorophyll Concentration	0.0014	0.684	0.002	0.998	−1.340	1.343
Sea Water Salinity	−0.3035	0.704	−0.431	0.666	−1.683	1.076

## Data Availability

The data presented in this study are available on request from the corresponding author due to ethical restrictions regarding participant privacy and confidentiality.

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
