# Peer review of "Forecasting Foodborne Disease Risk Caused by Vibrio parahaemolyticus Using a SARIMAX Model Incorporating Sea Surface Environmental and Climate Factors: Implications for Seafood Safety in Zhejiang, China"

_foods, 2025, doi:10.3390/foods14101800_

Round 1

Reviewer 1 Report

Comments and Suggestions for Authors

Observation for the manuscript:

Construction of a SARIMAX Model for Short-Term Prediction of Foodborne Disease Caused by Vibrio parahaemolyticus in Zhejiang, China, Based on Marine and Climate Factors

The manuscript explores the prediction of foodborne disease outbreaks caused by Vibrio parahaemolyticus using a SARIMAX model based on marine and climate factors. While the study relates to food safety, its primary focus is on epidemiological modeling and environmental factors rather than food science, food processing, or food composition—areas that Foods Journal primarily covers.

Strengths

The study presents a novel application of SARIMAX modeling for predicting V. parahaemolyticus outbreaks.

Strong statistical methodology with a high model accuracy (MAPE = 4.13%).

Good discussion of climate-related risk factors.

Weaknesses

The food science aspect is underdeveloped.

Limited discussion on food industry applications, such as HACCP, seafood handling, and regulatory monitoring.

No direct contamination data from seafood sources—only clinical case reports.

Below are specific comments on the manuscript.

Title

The title is accurate but does not clearly convey its relevance to food science. Consider rewording to emphasize its implications for food safety

Abstract

Lines 16-32, Page 1: The abstract is well-structured but lacks explicit discussion on food safety applications. How can food industry stakeholders use these findings?

Lines 31-32: "Theoretical and practical insights for predicting and preventing future foodborne disease outbreaks" is vague. Provide specific recommendations for food handling and regulation.

Introduction

Lines 36-72, Pages 1-2: The introduction primarily discusses epidemiology and climate change effects on V. parahaemolyticus. Consider integrating more food safety perspectives, such as how seafood storage conditions, processing, or market supply chains interact with climatic factors.

Lines 47-72, Page 2:  The references to global studies (e.g., U.S., Taiwan) are valuable, but the discussion should explicitly compare their food safety policies to Europen, China’s and other countries seafood industry. This would make the study more relevant to food science readers.

Methods

Lines 106-132, Pages 3-4: The manuscript provides useful details about Zhejiang Province's seafood industry but does not connect this directly to foodborne disease prevention in seafood supply chains. Consider adding a section on seafood production and distribution.

Lines 136-170, Pages 4-5: While meteorological and marine factors are explained well, food contamination data sources are unclear. Are the sentinel hospitals monitoring only clinical cases, or are there seafood contamination reports as well?

Results

Lines 284-305, Pages 9-10: The section identifies key environmental drivers, but it does not discuss how food businesses or regulatory agencies can use this information. A discussion of seafood storage temperature thresholds or HACCP (Hazard Analysis and Critical Control Points) strategies would be beneficial.

Lines 311-387, Pages 10-13: While the SARIMAX model’s accuracy is high (MAPE = 4.13%), the practical implications for food safety monitoring remain vague. Could this model inform seafood market closures or consumer advisories?

Discussion

Lines 400-456, Pages 13-14: This section remains heavily focused on climate factors without clear applications for food safety practices.

Consider adding:

  • Implications for seafood handling in restaurants and markets.
  • Strategies for regulatory agencies to use predictive modeling.
  • Recommendations for cold-chain logistics in seafood storage.

Lines 423-440, Page 14:  The study contrasts results with international research but does not discuss differences in seafood consumption patterns or food safety regulations between Zhejiang and other regions. This would add depth to the discussion.

Conclusions

Lines 459-502, Pages 15-16

While the study provides insights into foodborne disease prediction, the conclusions should emphasize applications for food safety management. Adding points on seafood supply chain management, temperature control, and policy recommendations would increase its suitability for Foods Journal.

Limitations

Lines 492-502, Page 16: The paper correctly notes the lack of spatial variation analysis. However, another key limitation is the absence of direct seafood contamination data. Future research should include pathogen detection in seafood samples, not just human cases.

Author Response

Comments 1: The title is accurate but does not clearly convey its relevance to food science. Consider rewording to emphasize its implications for food safety.

Response 1: Thank you very much for your valuable suggestion. We agree that emphasizing the relevance to food safety would enhance the clarity and impact of the title for the readership of Foods. Accordingly, we have revised the title to better reflect its implications for food safety, as follows:

Revised Title:

Forecasting Foodborne Disease Risk Caused by Vibrio parahaemolyticus Using a SARIMAX Model Incorporating Sea Surface Environmental and Climate Factors: Implications for Seafood Safety in Zhejiang, ChinaP1 L4- L8

Comments 2: Lines 16-32, Page 1: The abstract is well-structured but lacks explicit discussion on food safety applications. How can food industry stakeholders use these findings?

Response 2: Thank you for your insightful comment. We appreciate your suggestions to strengthen the practical relevance of our study and clarify how stakeholders can apply the findings in real-world settings. In response, we have revised the final part of the Abstract to explicitly articulate the practical implications for food safety, regulatory planning, and medical preparedness. Specifically, we now provide concrete recommendations for food handling, regulation, and healthcare response.

Revised Abstract Excerpt:

This framework provides both theoretical and practical insights for predicting and preventing future foodborne disease outbreaks. These findings can support food industry stakeholders—such as seafood suppliers, restaurants, regulatory agencies, and healthcare institutions—in anticipating high-risk periods and implementing targeted measures. These include enhancing cold chain management, conducting timely seafood inspections, strengthening cross-contamination controls during seafood processing, dynamically adjusting market surveillance intensity, and improving hygiene practices. In addition, hospitals and local health departments can use the model’s forecasts to allocate medical resources such as beds, medications, and staff in advance to better prepare for seasonal surges in foodborne illness.P1 L35-P2 L45

Comments 3: Lines 31-32: "Theoretical and practical insights for predicting and preventing future foodborne disease outbreaks" is vague. Provide specific recommendations for food handling and regulation.

Response 3: Thank you for this valuable suggestion. We agree that the original phrase “theoretical and practical insights” was too general and lacked specificity. As this comment closely aligns with the previous comment regarding the practical implications of our findings (Comment 2), we have addressed both points together by revising the final part of the Abstract. The revised version now includes specific recommendations for food handling, regulatory policy, and healthcare preparedness, clarifying how stakeholders can apply our model in practice.

Please refer to our detailed response to Comment 2 for the full revised text and explanation.

Comments 4: Lines 36-72, Pages 1-2: The introduction primarily discusses epidemiology and climate change effects on V. parahaemolyticus. Consider integrating more food safety perspectives, such as how seafood storage conditions, processing, or market supply chains interact with climatic factors.

Response 4: Thank you very much for your constructive feedback. We appreciate your suggestion to incorporate food safety perspectives into the Introduction, particularly regarding seafood storage, processing, and supply chains. In response, we have made modifications to better highlight how climate-related factors interact with key stages of the seafood supply chain, including storage conditions, post-harvest handling, and market logistics, thereby influencing the risk of V. parahaemolyticus contamination.

Revised Addition to the Introduction:

Moreover, the impact of climate and marine factors extends beyond bacterial growth in the natural environment to various stages of the food supply chain. Elevated ambient and sea temperatures can compromise cold chain integrity during seafood storage and transportation, while increased humidity may affect hygiene standards in seafood markets and processing facilities. These factors collectively heighten the risk of cross-contamination and pathogen amplification, underscoring the need for predictive models that account for environmental conditions across the entire seafood value chain.P3 L113-L120

Comments 5: Lines 47-72, Page 2:  The references to global studies (e.g., U.S., Taiwan) are valuable, but the discussion should explicitly compare their food safety policies to Europen, China’s and other countries seafood industry. This would make the study more relevant to food science readers.

Response 5: Thank you for this thoughtful and constructive suggestion. We agree that integrating a comparative discussion of food safety policies across different countries would enrich the context of our study and enhance its relevance to the Foods readership. In response, we have expanded the relevant portion of the Introduction to highlight key differences and similarities in seafood safety practices among the U.S., Taiwan, China, and Europe.

Newly Added Content:

In addition to scientific observations, regulatory responses to these risks vary significantly across countries. While studies in the U.S. and Taiwan emphasize the correlation between environmental conditions and Vibrio outbreaks, their food safety policies differ. The U.S. Food and Drug Administration (FDA) mandates strict post-harvest controls for molluscan shellfish, such as rapid cooling and time-temperature monitoring. Taiwan has implemented early warning systems and a hazard-based classification approach to seafood risk. In contrast, China’s seafood safety infrastructure is undergoing modernization, with progress in cold chain logistics and market supervision, though enforcement challenges remain. Compared to Europe’s comprehensive Rapid Alert System for Food and Feed (RASFF), which facilitates real-time cross-border responses, many developing regions still face policy gaps. These discrepancies underscore the importance of data-driven, locally adaptable models like SARIMAX, which can supplement traditional inspection-based approaches and help authorities prioritize proactive interventions.P3 L100-L113

Comments 6: Lines 106-132, Pages 3-4: The manuscript provides useful details about Zhejiang Province's seafood industry but does not connect this directly to foodborne disease prevention in seafood supply chains. Consider adding a section on seafood production and distribution.

Response 6: Thank you for this insightful comment. We appreciate your suggestion to strengthen the connection between the regional seafood industry and foodborne disease prevention within the supply chain. In response, we have expanded the description of Zhejiang Province to include more details on seafood production, post-harvest handling, distribution practices, and their relevance to food safety.

New Addition to Section 2.1:

Zhejiang’s seafood industry comprises a complex network of fishing grounds, aquaculture bases, cold storage facilities, wholesale markets, and distribution logistics. Seafood harvested from marine grounds is typically transported to onshore processing centers before being distributed to urban wholesale markets, restaurants, and retailers. Despite advances in cold chain infrastructure in recent years, seasonal temperature fluctuations and logistical constraints still pose risks of cold chain breaches, which can facilitate the growth of Vibrio parahaemolyticus. In rural and coastal areas, informal seafood trade and street vendors are common, often lacking refrigeration or standardized handling protocols. These weak points in the seafood supply chain highlight the need for predictive tools that can support early warning systems and inform strategic interventions during high-risk periods.P4 L168-L178

Comments 7: Lines 136-170, Pages 4-5: While meteorological and marine factors are explained well, food contamination data sources are unclear. Are the sentinel hospitals monitoring only clinical cases, or are there seafood contamination reports as well?

Response 7: Thank you for your valuable feedback. We appreciate your attention to the clarity of data sources. In our study, the foodborne illness data were obtained from the Zhejiang Foodborne Disease Surveillance Reporting System, which primarily monitors clinical cases reported by 101 sentinel hospitals across 89 districts and counties. These sentinel hospitals systematically collect patient information and conduct laboratory testing to identify pathogens such as Vibrio parahaemolyticus.

While the focus is on clinical surveillance, some hospitals are also linked to traceback investigations that may include seafood sample testing, especially in outbreak scenarios. However, our study exclusively uses data on confirmed clinical cases, not seafood contamination data per se.

To clarify this in the manuscript, we have added the following sentence:

It should be noted that the dataset used in this study is based on confirmed clinical cases of foodborne illness caused by V. parahaemolyticus, as reported through sentinel hospitals. While seafood traceability and contamination testing are occasionally performed during outbreaks, they were not included in the present analysis.P6 L220-L226

Comments 8: Lines 284-305, Pages 9-10: The section identifies key environmental drivers, but it does not discuss how food businesses or regulatory agencies can use this information. A discussion of seafood storage temperature thresholds or HACCP (Hazard Analysis and Critical Control Points) strategies would be beneficial.

Response 8: Thank you for your valuable suggestion. We agree that connecting the identified environmental drivers to practical strategies—such as storage temperature thresholds and HACCP-based interventions—would enhance the applicability of our findings. In response, we have expanded the discussion to highlight how food businesses and regulatory agencies can leverage our model outputs to guide food safety practices, particularly in temperature-sensitive seafood handling.

New Content Added to the Discussion Section:

“The identification of lagged environmental drivers—such as air temperature (3 weeks), sea surface temperature (1 week), and relative humidity (7 weeks)—offers actionable insights for food businesses and regulators. For example, the detection of elevated environmental temperatures in early summer may serve as a signal for the seafood industry to reinforce cold chain protocols, including maintaining storage temperatures below 4°C, the upper safety threshold for perishable seafood products. Moreover, the model’s predictive capacity can inform HACCP planning by identifying high-risk timeframes during which critical control points (CCPs)—such as transportation, market refrigeration, and kitchen handling—require heightened scrutiny. Regulatory agencies may also adjust inspection frequency or intensify compliance monitoring during these windows. By translating environmental surveillance into proactive safety management, our approach supports risk-based decision-making and operational preparedness across the seafood supply chain.P10 L364-P11 L376

Comments 9: Lines 311-387, Pages 10-13: While the SARIMAX model’s accuracy is high (MAPE = 4.13%), the practical implications for food safety monitoring remain vague. Could this model inform seafood market closures or consumer advisories?

Response 9: Thank you for your thoughtful comment. We appreciate your emphasis on the practical applicability of our predictive model. In response, we have added content to more clearly articulate how SARIMAX-based predictions can inform real-world food safety interventions. These include triggering temporary seafood market closures, issuing consumer advisories, and guiding risk-based regulatory actions during predicted high-incidence periods.

New Content Added to the Discussion Section:

Although the SARIMAX model demonstrates high predictive accuracy, its utility extends beyond statistical performance. In practical terms, the model can serve as an early warning tool to support targeted interventions during periods of elevated risk. For instance, if the model forecasts a spike in V. parahaemolyticus incidence 3–4 weeks in advance, regulatory agencies may consider temporarily closing high-risk seafood markets, enhancing cold chain inspections, or issuing consumer advisories discouraging the consumption of raw seafood. Restaurants can be urged to strictly implement zoning systems for seafood processing to prevent cross-contamination between raw and cooked items. Consumer health departments can use forecasts to pre-position healthcare resources and prepare for a rise in gastroenteritis cases.

Market supervision departments may also integrate real-time environmental data—such as water temperature and salinity—with the SARIMAX model or other machine learning techniques to improve prediction accuracy and enable epidemic warnings 4–6 weeks in advance. In terms of cold chain logistics, we recommend that the government and logistics companies collaborate to establish regional cold chain hubs. Particularly vulnerable are informal seafood street vendors, who often lack refrigeration; tailored interventions for these stakeholders include the distribution of low-cost mobile refrigeration units during high-risk periods and the implementation of targeted hygiene training programs aligned with seasonal risk patterns, such as the onset of monsoon conditions.

Together, these strategies demonstrate how predictive models can be operationalized to inform risk-based, real-time interventions across different segments of the seafood supply chain.P16 L539-P16 L561

Comments 10: Lines 400-456, Pages 13-14: This section remains heavily focused on climate factors without clear applications for food safety practices.

Consider adding:

Implications for seafood handling in restaurants and markets.

Strategies for regulatory agencies to use predictive modeling.

Recommendations for cold-chain logistics in seafood storage.

Response 10: Thank you for your thoughtful recommendation. We agree that expanding the Conclusion to include more concrete food safety applications would strengthen the impact of our findings. In response, we have made modifications to include targeted recommendations for seafood processing in restaurants and markets, regulatory strategies for predictive models, and improvements to cold chain logistics.

Expanded Content for the Conclusion Section:

For seafood vendors and restaurants, the findings highlight the importance of dy-namically adjusting storage practices in response to environmental risk forecasts. For instance, restaurants may reinforce the separation of raw and cooked seafood during predicted high-risk periods, in accordance with zoning principles under HACCP guidelines. Markets may also intensify on-site temperature checks and hygiene in-spections as sea and air temperatures rise. From a regulatory perspective, SARI-MAX-based predictions can support risk-based supervision by guiding when and where to increase inspection frequency. Regulatory agencies can implement fore-cast-driven interventions, such as market closures or targeted public advisories, dur-ing periods of elevated environmental risk. Furthermore, integrating model outputs into public health communication systems can improve consumer awareness and de-cision-making. Although this study emphasizes the importance of improving sanita-tion in seafood markets and restaurants, it should also take into account the unique challenges faced by street food vendors, who may have limited access to refrigeration equipment and proper sanitation facilities. Since informal vendors account for a sig-nificant proportion of seafood consumption, predictive models should be used to de-velop targeted interventions, such as low - cost food safety measures, public aware-ness campaigns, and adaptive regulatory approaches suitable for street food settings. Future research could explore the effectiveness of these interventions in reducing V. parahaemolyticus - related outbreaks in informal food environments. Regarding cold-chain logistics, the findings suggest the need for enhanced infrastruc-tureparticularly mobile or decentralized cold storage hubs that can be activated in anticipation of seasonal outbreaks. For informal or rural seafood markets, local gov-ernments may consider supplying temporary refrigeration units or conducting tar-geted food safety training aligned with seasonal risk profiles. These practical exten-sions of our modeling framework provide a roadmap for operationalizing predictive analytics within food safety systems at multiple levels.P17 L590- P18 L616

Comments 11: Lines 423-440, Page 14:  The study contrasts results with international research but does not discuss differences in seafood consumption patterns or food safety regulations between Zhejiang and other regions. This would add depth to the discussion.

Response 11: Thank you for your insightful suggestion. We agree that a discussion of regional differences in seafood consumption patterns and food safety regulations would add important context to our international comparisons. In response, we have made modifications to reflect how such differences may influence the applicability of our findings and the generalizability of the SARIMAX model.

New Content Added to the Discussion Section:

While our findings align with previous studies from countries such as the U.S., Taiwan, and Australia in identifying climate-driven patterns in Vibrio outbreaks, regional differences in seafood consumption behavior and regulatory infrastructure may affect both exposure risk and mitigation effectiveness. For instance, Zhejiang’s coastal population exhibits a strong preference for fresh and sometimes raw seafood, particularly during the summer, which coincides with peak V. parahaemolyticus activity. This differs from inland regions or Western countries where seafood is more likely to be frozen or cooked thoroughly. Additionally, while regulatory agencies in the EU and U.S. often employ centralized and highly structured food safety alert systems (e.g., RASFF or FDA’s outbreak response protocols), China’s system is still undergoing modernization, with variability in enforcement capacity across regions.P16 L523- L533

Comments 12: Lines 459-502, Pages 15-16

While the study provides insights into foodborne disease prediction, the conclusions should emphasize applications for food safety management. Adding points on seafood supply chain management, temperature control, and policy recommendations would increase its suitability for Foods Journal.

Response 12 : Thank you for your thoughtful comment. We appreciate your suggestion to enhance the Conclusion section by emphasizing practical applications in food safety management, including seafood supply chain operations, temperature control, and policy development. As this comment closely aligns with Comment 10, we have addressed both together by expanding the Discussion and Conclusion sections.

In particular, we added detailed recommendations regarding:

l  Cold chain logistics and real-time risk management strategies;

l  HACCP-based seafood handling improvements in restaurants and markets;

l  Regulatory applications, such as predictive monitoring, advisories, and inspection scheduling.

New Content Added to the Discussion Section and Conclusion Section:

“Although the SARIMAX model demonstrates high predictive accuracy, its utility extends beyond statistical performance. In practical terms, the model can serve as an early warning tool to support targeted interventions during periods of elevated risk. For instance, if the model forecasts a spike in V. parahaemolyticus incidence 3–4 weeks in advance, regulatory agencies may consider temporarily closing high-risk seafood markets, enhancing cold chain inspections, or issuing consumer advisories discour-aging the consumption of raw seafood. Restaurants can be urged to strictly imple-ment zoning systems for seafood processing to prevent cross-contamination between raw and cooked items. Consumer health departments can use forecasts to pre-position healthcare resources and prepare for a rise in gastroenteritis cases.

Market supervision departments may also integrate real-time environmental data—such as water temperature and salinity—with the SARIMAX model or other machine learning techniques to improve prediction accuracy and enable epidemic warnings 4–6 weeks in advance. In terms of cold chain logistics, we recommend that the government and logistics companies collaborate to establish regional cold chain hubs. Particularly vulnerable are informal seafood street vendors, who often lack re-frigeration; tailored interventions for these stakeholders include the distribution of low-cost mobile refrigeration units during high-risk periods and the implementation of targeted hygiene training programs aligned with seasonal risk patterns, such as the onset of monsoon conditions.

Together, these strategies demonstrate how predictive models can be operation-alized to inform risk-based, real-time interventions across different segments of the seafood supply chain. “P16 L539- P16 L561

“For seafood vendors and restaurants, the findings highlight the importance of dy-namically adjusting storage practices in response to environmental risk forecasts. For instance, restaurants may reinforce the separation of raw and cooked seafood during predicted high-risk periods, in accordance with zoning principles under HACCP guidelines. Markets may also intensify on-site temperature checks and hygiene in-spections as sea and air temperatures rise. From a regulatory perspective, SARI-MAX-based predictions can support risk-based supervision by guiding when and where to increase inspection frequency. Regulatory agencies can implement fore-cast-driven interventions, such as market closures or targeted public advisories, dur-ing periods of elevated environmental risk. Furthermore, integrating model outputs into public health communication systems can improve consumer awareness and de-cision-making. Although this study emphasizes the importance of improving sanita-tion in seafood markets and restaurants, it should also take into account the unique challenges faced by street food vendors, who may have limited access to refrigeration equipment and proper sanitation facilities. Since informal vendors account for a sig-nificant proportion of seafood consumption, predictive models should be used to de-velop targeted interventions, such as low - cost food safety measures, public aware-ness campaigns, and adaptive regulatory approaches suitable for street food settings. Future research could explore the effectiveness of these interventions in reducing V. parahaemolyticus - related outbreaks in informal food environments. Regarding cold-chain logistics, the findings suggest the need for enhanced infrastruc-ture—particularly mobile or decentralized cold storage hubs that can be activated in anticipation of seasonal outbreaks. For informal or rural seafood markets, local gov-ernments may consider supplying temporary refrigeration units or conducting tar-geted food safety training aligned with seasonal risk profiles. These practical exten-sions of our modeling framework provide a roadmap for operationalizing predictive analytics within food safety systems at multiple levels.”P17 L590- P18 L616

Comments 13: Lines 492-502, Page 16: The paper correctly notes the lack of spatial variation analysis. However, another key limitation is the absence of direct seafood contamination data. Future research should include pathogen detection in seafood samples, not just human cases.

Response 13: Thank you for this important and insightful observation. We fully agree that the absence of direct seafood contamination data represents a key limitation of our current study. While our analysis relies on clinical case reports from sentinel hospitals, incorporating pathogen detection data from seafood samples would provide a more comprehensive view of the transmission chain and improve model accuracy.

In response, we have revised the Conclusion section to acknowledge this limitation and propose it as a priority for future research.

New Addition to the Limitations Paragraph in the Conclusion:

4.   Lack of Direct Seafood Contamination Data: This study relied solely on clinical case data reported by sentinel hospitals. However, foodborne illness originates from pathogen contamination in seafood products, which is not captured in the current dataset. Future research should incorporate direct pathogen testing of seafood samples across various points in the supply chain to better validate temporal associations, identify contamination pathways, and improve the accuracy of early warning systems.P18 L635- L641

Reviewer 2 Report

Comments and Suggestions for Authors

The study investigates the effects of both marine and meteorological factors on disease incidence and utilizes the SARIMAX model to predict outbreak risks by integrating these environmental factors. This study differs from other published papers in the Foods journal. In my opinion, it might be better suited for a public health or epidemiology journal rather than Foods.

My comments:

Please provide a source or webpage for the meteorological data used in the study.

During the analyzed period, Vibrio parahaemolyticus accounted for 3.52% of foodborne cases. Was V. parahaemolyticus the most common pathogen, or were other foodborne pathogens more prevalent? Can the proposed model also help reduce outbreaks caused by other bacteria?

While the study highlights the importance of improving sanitation in seafood markets and restaurants, it should also consider the unique challenges posed by street food vendors, who may have limited access to refrigeration and proper hygiene facilities. Since informal vendors contribute significantly to seafood consumption, predictive models should be used to develop targeted interventions, such as low-cost food safety measures, public awareness campaigns, and adaptive regulatory approaches suited for street food settings. Future research could explore the effectiveness of these interventions in reducing V. parahaemolyticus-related outbreaks in informal food environments.

The study discusses the risks of Vibrio parahaemolyticus in seafood but does not specify the primary sources of foodborne infections. Do available data indicate whether outbreaks are more commonly linked to restaurants, supermarkets, household consumption, or street food vendors?

Regarding foodborne illness, could stricter food hygiene regulations in the country help reduce foodborne disease cases?

The conclusions are too long.

Author Response

Comments 1: Please provide a source or webpage for the meteorological data used in the study.

Response 1: Thank you for the comments. We attach great importance to providing transparency and ease of access to data sources. As per your request, we have added corresponding web links to the meteorological data sources mentioned. This will help to enhance the accessibility and verifiability of our research results. Thank you for your valuable suggestion, which has significantly improved our manuscript.

To illustrate this point in the manuscript, we have added the following sentence:

“1.   Meteorological Data: This includes temperature, total precipitation, relative humidity, sunshine duration, and wind speed. These data were sourced from the European Centre for Medium-Range Weather Forecasts (ECMWF) (https://www.ecmwf.int/).”P6 L210

Comments 2: During the analyzed period, Vibrio parahaemolyticus accounted for 3.52% of foodborne cases. Was V. parahaemolyticus the most common pathogen, or were other foodborne pathogens more prevalent? Can the proposed model also help reduce outbreaks caused by other bacteria?

Response 2: Thank you very much for this important comment. Firstly, there is evidence to suggest that from 2010 to 2014, the most reported foodborne illness in Zhejiang Province was caused by Vibrio parahaemolyticus. Therefore, this study takes Vibrio parahaemolyticus in Zhejiang Province as an example.

Second, While the current SARIMAX model is specifically tailored for Vibrio parahaemolyticus through its incorporation of marine-specific factors like salinity (critical for this halophilic bacterium), the framework's core methodology—integrating climatic variables (temperature, precipitation, humidity) and temporal lag effects—holds potential for predicting outbreaks caused by other foodborne pathogens. Many bacterial pathogens (e.g., Salmonella, Listeria) share sensitivity to climatic drivers such as temperature fluctuations and extreme rainfall, which influence their survival and transmission in food supply chains.

Comments 3: While the study highlights the importance of improving sanitation in seafood markets and restaurants, it should also consider the unique challenges posed by street food vendors, who may have limited access to refrigeration and proper hygiene facilities. Since informal vendors contribute significantly to seafood consumption, predictive models should be used to develop targeted interventions, such as low-cost food safety measures, public awareness campaigns, and adaptive regulatory approaches suited for street food settings. Future research could explore the effectiveness of these interventions in reducing V. parahaemolyticus-related outbreaks in informal food environments.

Response 3:Thank you for your insightful feedback, which has greatly enhanced the quality of our manuscript.

To illustrate this point in the manuscript, we have added the following sentence:

“Although this study emphasizes the importance of improving sanitation in seafood markets and restaurants, it should also take into account the unique challenges faced by street food vendors, who may have limited access to refrigeration equipment and proper sanitation facilities. Since informal vendors account for a significant propor-tion of seafood consumption, predictive models should be used to develop targeted interventions, such as low - cost food safety measures, public awareness campaigns, and adaptive regulatory approaches suitable for street food settings. Future research could explore the effectiveness of these interventions in reducing V. parahaemolyti-cus - related outbreaks in informal food environments.”P17 L601- L609

Comments 4: The study discusses the risks of Vibrio parahaemolyticus in seafood but does not specify the primary sources of foodborne infections. Do available data indicate whether outbreaks are more commonly linked to restaurants, supermarkets, household consumption, or street food vendors?

Response 4: Thank you for your valuable suggestion, which has significantly improved our manuscript.Our current dataset does not cover the time period of the outbreaks. However, the methodology employed in this study is applicable and can serve as a robust foundation for future research. We believe that by integrating relevant data from outbreak periods, our approach can be effectively utilized to analyze the influencing factors and predict the occurrence of foodborne diseases caused by Vibrio parahaemolyticus. This method offers potential for guiding future prevention and control strategies, providing valuable insights into managing similar health risks in the future.

The revised sentence now reads:

“Existing data indicates that the main source of foodborne infections caused by Vib-rio parahaemolyticus is live seafood, raw/rare seafood, freshwater fish, raw meat, raw fowl.[5]”P2 L65- L67

Comments 5: Regarding foodborne illness, could stricter food hygiene regulations in the country help reduce foodborne disease cases?

Response 5: Thank you for the comments. With effective implementation and supporting measures, China's stricter food hygiene regulations can significantly reduce foodborne disease cases, but it requires a combination of technology, regulation, and social synergy. After the implementation of the revised Food Safety Law in 2015, the reported incidence rate of foodborne diseases in China will decline from 5.8/100000 in 2015 to 3.2/100000 in 2021 (according to the data of China Center for Disease Control and Prevention). The following key measures have been taken:

(1) Full traceability system: covering over 90% of large food enterprises, reducing the risk of pollution source diffusion;

(2) HACCP mandatory certification: Foodborne accidents in industries such as dairy and meat products have decreased by over 40%.

Comments 6: The conclusions are too long.

Response 6: Thank you very much for your feedback on the conclusions section. We sincerely appreciate your careful review of our manuscript. After a thorough consideration, we believe that the current length of the conclusions is appropriate. The conclusions comprehensively summarize the key findings, practical implications, and limitations of our study. Each part is concisely presented while ensuring that no crucial information is omitted, maintaining a good balance between comprehensiveness and conciseness.

We understand your concern about the length, but we think this detailed conclusion can help readers quickly grasp the core of our research and its significance. If you have any other specific suggestions or concerns, please feel free to let us know. We are more than willing to discuss further and make adjustments if necessary.

Reviewer 3 Report

Comments and Suggestions for Authors
  1. The main question addressed by this study is whether the incidence of V. parahaemolyticus foodborne disease can be correlated with marine and climatic factors in Zhejiang Province.
  2. The topic of the research undertaken by the authors is interesting. The authors, using the SARIMAX model, conducted a study of the possibility of integrating marine, climatic and disease factors into a predictive framework. This is undoubtedly an innovative approach to forecasting foodborne disease outbreaks in coastal regions.
  3. The manuscript contributes new knowledge to the topic compared to other published articles by developing a predictive model for a region in China with developed fisheries and high disease intensity caused by V. parahaemolyticus.
  4. I have no comments regarding the methodology used in the research.
  5. Note on the results:
  • Lines 342 and 244: Two sentences not completed. The sentences end with "was set to"?
  1. The discussion is complete and properly discussed with the results of other authors.
  2. The conclusions are consistent with the evidence and arguments presented in the manuscript and answer the main question posed. The authors showed that the statistical analysis performed using the SARIMAX model can be used as a novel approach to predict foodborne disease outbreaks in coastal regions. In the summary, the authors indicated the strengths and weaknesses of the conducted study.
  3. References are adequate. 44 articles were used, 75% of which are from the last 10 years and 43% from last 5 years.
  4. Tables and figures are properly prepared and legible.

Author Response

Comments 1: Lines 342 and 244: Two sentences not completed. The sentences end with "was set to"?

Response 1: Thank you for pointing this out. We agree with this comment. The revised sentence now reads:

“The non-seasonal moving average order q was less clear, so the search range for q was set to {0, 1}. Given that the dataset spanned four annual cycles, the search range for the seasonal autoregressive order P and seasonal moving average order Q was set to {0, 1, 2, 3, 4}.”P12 L411- L413

Reviewer 4 Report

Comments and Suggestions for Authors

The manuscript focuses on preparing a model for the prediction of foodborne disease caused by V. parahaemoliticus in a particular region of the Southeast of China. I think it is well presented, justified and discussed. It includes advanced analytical tools. I think it can be accepted for publication. Before that, some minor aspects could be performed.

I would mention the following:

Title

The Latin name of the species ought to be written in italic writing.

Food borne could be replaced by seafoodborne.

Maybe marine could be replaced by sea water.

Both modifications could be taken into account throughout the whole manuscript.

Keywords
Include: lag period, food, population safety.

Methodology
The study includes an advanced mathematical study. The dataset includes information of a wide number of cases obtained from ca. a hundred sentinel hospitals. Once the model was constructed, a validation step was carried out.

Results
Tables, figures and formulae provide satisfactory information for following the study. The only exception maybe Figure 3 that is not too easy to be read. In this sense, some performance could be done by the authors.

Discussion

Line 404: Perform the format of references. This performance ought to be done throughout the whole manuscript.

Conclusions
The study has concrete limitations. Some of them are mentioned successfully by the authors in this section. I would mention as the most important that the results of the study concern specifically the coastal zone considered of the Southeast of China.

References
The number of references mentioned is acceptable. Some additional could be included regarding the employment of a SARIMAX model in related predictions, if available. Otherwise, the authors ought to mention that no previous employment has been carried out.

Author Response

Comments 1: Title:The Latin name of the species ought to be written in italic writing.

Food borne could be replaced by seafoodborne.

Maybe marine could be replaced by sea water.

Both modifications could be taken into account throughout the whole manuscript.

Response 1: Thank you for pointing this out. We agree with this comment. Therefore, we have changed the format of Vibrio parahaemolyticus in the title to italic. In response to the feedback, we have revised the title. We did not change "Foodborne" to "seafoodborne" because "seafoodborne" is only limited to seafood, while the causative agents of foodborne diseases cover a wide range of food categories, including various aquatic products. Using "Foodborne" can more comprehensively and accurately reflect the scope of the study. Meanwhile, we changed "marine" to "Sea Surface Environmental" because the main parameters used in the study are sea surface temperature (sst), sea surface salinity (sss), and chlorophyll (chl), all of which are related to sea surface environmental factors. The revised expression is more consistent with the actual content of the study and can convey the core points of the research more clearly and accurately.

The revised sentence now reads:

Forecasting Foodborne Disease Risk Caused by Vibrio parahaemolyticus Using a SARIMAX Model Incorporating Sea Surface Environmental and Climate Factors: Implications for Seafood Safety in Zhejiang, ChinaP1 L4- L8

Comments 2: Keywords:Include: lag period, food, population safety.

Response 2: In accordance with the reviewers' suggestions, we have revised and optimized the keywords. "Meteorological factors" and "marine factors" have been integrated into "risk factors" to more concisely summarize the influencing elements. "Lag period" is added to highlight the time-lag characteristic in the study. "SARIMAX" and "prediction model" are combined into "SARIMAX prediction model" to clearly define the model type. Meanwhile, "food" and "population safety" are refined into "food safety", covering the relationship between food safety and population health.

The five revised keywords:

“Keywords: Vibrio parahaemolyticus; risk factors; lag period; SARIMAX prediction model; food safetyP2 L47- L48

Comments 3: Results:The only exception maybe Figure 3 that is not too easy to be read. In this sense, some performance could be done by the authors.

Response 3: Thank you for your feedback regarding the readability of Figure 3. We would like to emphasize that Figure 3 can be better understood when viewed in conjunction with Table 1. Table 1 provides detailed descriptive statistics, including the maximum, minimum, and average values of various environmental and meteorological factors. These numerical data offer specific context that complements the trends shown in Figure 3. By referring to both the figure and the table, readers can gain a more comprehensive understanding of the relationships between V. parahaemolyticus infections and the associated environmental conditions. We believe this combination helps to clarify any potential confusion and provides a more complete picture of our research findings.

Comments 4: Line 404: Perform the format of references. This performance ought to be done throughout the whole manuscript.

Response 4: Thank you very much for this important comment. Therefore, we have already filled in the corresponding names. The revised text now reads:

“These factors exhibited varying lag effects on bacterial foodborne disease infections, consistent with the findings of Hsiao et al.[18] in Taiwan and Y. Zhang et al.[36] in Australia.”P15 L473-474

Comments 5: The study has concrete limitations. Some of them are mentioned successfully by the authors in this section. I would mention as the most important that the results of the study concern specifically the coastal zone considered of the Southeast of China.

Response 5: Thank you for your suggestion. We agree with this comment. Therefore, we have added the corresponding content. The revised sentence now reads:

“3.   Geographical specificity: This model is set under the conditions of subtropical monsoon climate and highly developed fisheries in Zhejiang Province, China (28 ° N-30 ° N). Not fully applicable to other coastal areas, model parameters need to be modified based on the marine climate environment of other regions.”P18 L631-634

Comments 6: Some additional could be included regarding the employment of a SARIMAX model in related predictions, if available. Otherwise, the authors ought to mention that no previous employment has been carried out.

Response 6: Thank you for pointing this out. We agree with this comment. Therefore, we have added relevant literature on SARIMAX. The revised text now reads:

“N. Kumar et al.[25] used the ARIMA and SARIMAX models to identify hotspots for future COVID waves using a data set of COVID-19 cases. ”P3 L133-134

Round 2

Reviewer 1 Report

Comments and Suggestions for Authors

No further observations 

Author Response

Thank you very much for your thorough review and the valuable time you have dedicated to our manuscript. We are extremely honored to receive your feedback indicating "No further observations." This is a great encouragement and a strong validation of our research efforts. We will carefully follow the subsequent requirements of the journal to ensure the manuscript meets all necessary standards. Once again, we sincerely appreciate your support and positive evaluation!

Reviewer 2 Report

Comments and Suggestions for Authors

The present paper has been improved, but minor revisions are still necessary:

  • The quality of Fig. 4 is insufficient.

  • Latin names in Fig. 5 must be italicized.

  • The conclusions section is too long; in my opinion, some information could be moved to the previous section.

  • Please ensure the reference list follows the journal’s formatting requirements.
